# Learning a Distributed Hierarchical Locomotion Controller for Embodied Cooperation

**Chuye Hong**[1,*] **, Kangyao Huang**[1,*] **, Huaping Liu**[1,†]
[1]Department of Computer Science and Technology, Tsinghua University
[*]Equal contributions, [†]Corresponding author.

**Abstract:** In this work, we propose a distributed hierarchical locomotion control strategy for whole-body cooperation and demonstrate the potential for scaling to large numbers of agents. Our method utilizes a hierarchical structure to break down complex tasks into smaller, manageable sub-tasks. By incorporating spatiotemporal continuity features, we establish the sequential logic necessary for causal inference and cooperative behaviour in sequential tasks, thereby facilitating efficient and coordinated control strategies. Through training within this framework, we demonstrate enhanced adaptability and cooperation, leading to superior performance in task completion compared to the original methods. Moreover, we construct a set of environments as the benchmark for embodied cooperation. A visual depiction of highlights of the learned cooperative behaviours can be found on this website[1].

**Keywords:** cooperation, locomotion, hierarchical reinforcement learning

## 1 Introduction

Cooperatively accomplishing embodied tasks with multiple robots has consistently been a highly challenging area of research. Recent studies mainly focus on embodied manipulation cooperation among robotic arms or formation control over the upper level within a group of mobile robots [1, 2]. Nevertheless, multi-agent cooperation via whole-body and end-to-end locomotion control is rarely studied.

Some previous works showcase the manipulation via locomotion [3] but are only tested on two agent systems, and the scalability of this method is still agnostic for migration to any number of agent populations. In this work, we aim to realize more complex embodied multi-agent cooperation by learning a distributed hierarchical locomotion control system, decomposing the complex and coupled behaviours while maintaining the potential for unlimited expansion on the swarm. As the foundation for implementation and validation, we construct three scenarios in IsaacSim [4] as benchmarks for embodied cooperation study.

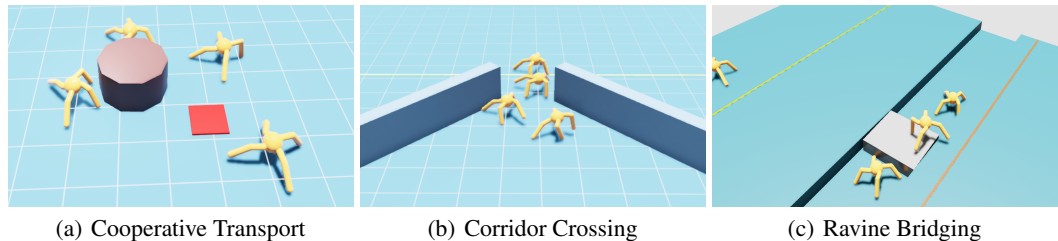

(a) Cooperative Transport  (b) Corridor Crossing  (c) Ravine Bridging

Figure 1: Embodied Cooperation Environments.

---

[1]Videos of our results are available at our project webpage: https://d-hrl.github.io

8th Conference on Robot Learning (CoRL 2024), Munich, Germany.

Concurrently, training a robot for a specific function can be effectively achieved through reinforcement learning (RL), such as learning movement patterns [5], interactive behaviours [6], as well as logical inference in games [7]. Although RL provides a recognized powerful exploration capability and tremendous progress has been made in sampling efficiency [4], finding and mastering a sequence of sophisticated tasks through searching remains a challenging problem. Hierarchical reinforcement learning (HRL) alleviates this to a certain extent, aiming to understand the logical relationships among "control, action, behaviour, dynamic outcomes, and feedback" in a segmented manner. An increasing number of studies are employing hierarchical approaches to decompose complex tasks [8, 9, 10, 11, 12, 13, 14, 15]. Inspired by these, we propose a fully-decentralized HRL framework to naturally decompose perception feature extraction, cooperative behaviours, and fundamental actions.

In the previous work [3], embodied cooperation is completed via locomotion control by a small number of robots. However, as the number of agents increases, previous methods fail to effectively address the issue, while the scalability of the swarm is crucial in multi-agent reinforcement learning (MARL) field [16, 17, 18, 19]. To tackle this challenge, we propose a fully-decentralized framework that can be applied to a flock of agents to enhance the method's scalability and promote impressive cooperative behavior among larger-scale intelligent agents. We aim to apply this framework to an expandable MAS. Moreover, spatiotemporal features between a sequence of events are not considered in the previous work, which causes failure in understanding interactive and cooperative behaviours. To enhance this, we deploy sequential behaviour learning with spatiotemporal memory capabilities in the middle layer of the hierarchical network by constructing a Recurrent Neural Network.

The key contributions of this work can be described as follows:

- A distributed HRL locomotion control framework is constructed for scale-expandable embodied cooperation. This architecture can be applied to both homogeneous and heterogeneous groups.
- We build a set of scenarios for whole-body embodied cooperation in IsaacSim shown in Fig. 1, as a benchmark for embodied cooperation.
- We showcase the remarkable emergence of cooperative behaviours among agents and demonstrate the advantages compared with previous approaches.

## 2 Related Works

### 2.1 Embodied Cooperation

Research on cooperation can be divided according to with or without body interaction. Some collaborative tasks are based on particle games, focusing solely on coordination and alignment at position/velocity levels [20, 21]. In contrast, *studies related to embodied cooperation lie on the foundations of the robotic platform that owns the real body, paying more attention to interaction, collision, and whole-body control.* Optimization methods and computational intelligence methods can solve embodied cooperation problems. The model predicted control (MPC) based methods are widely used in formation control on quadruped robots [22, 23, 24, 25] and aerial robots [26, 27], and nature-inspired algorithms can obtain good performance in a self-organized swarm cooperative task [2]. These approaches provide highly accurate control but lack flexibility and generalization. Learning-based methods can compensate for this [28, 3]. Moreover, the large language model (LLM) can provide robots with powerful cognitive reasoning capabilities for collaboration, bringing a new way to solve the problem [29, 30, 31, 32].

### 2.2 HRL in locomotion Control

HRL is well suited for complex and spatiotemporal tasks [33]. Its effectiveness has been proven and widely acknowledged [34]. Multiple steps can decouple an intricate task to reduce the training difficulty, corresponding to various layers of policies [35]. Decisions between layers can be made

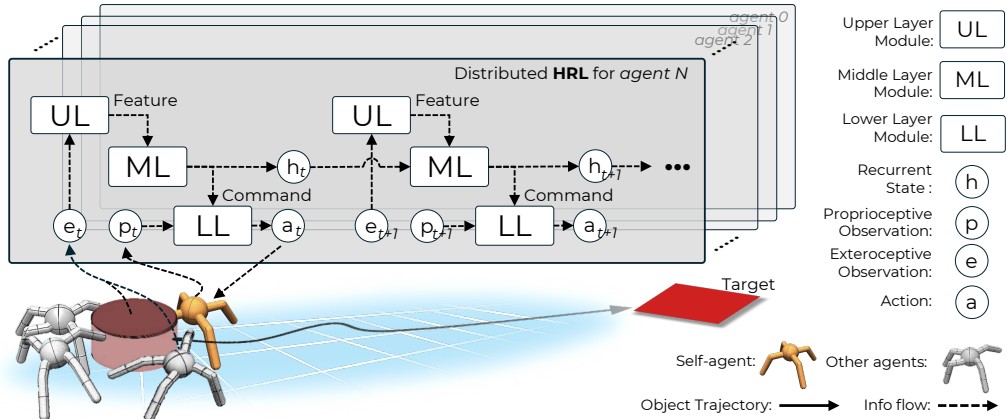

Figure 2: Distributed HRL pipeline: demonstrate the information flow within the distributed hierarchical reinforcement learning, using scenario Cooperative Transport as an example. Here we maintain a centralized training but distributed control hierarchical reinforcement learning framework, where we decompose the complex behaviours into three levels: Upper Layer (UL), Middle Layer (ML), and Lower Layer (LL). UL module processes the external information $e_t$ including environmental perception and relative position to colleagues, extracting features and sending to ML; ML module is a recurrent neural network layer that maintains a recurrent state $h$ considering temporal and spatial correlation, and outputs a locomotion command into LL module; LL module is a pre-trained locomotion control layer that generates action $a_t$ and applies it to agent according to the proprioceptive observation $p_t$, where we have two modes: position and velocity. The goal of Cooperative Transport is to move a cylinder object to the red target zone collaboratively by a group of Ant robots.

by deciding whether the previous action has been completed [36]. Essentially, hierarchical learning provides artificial direction and reduces the need for model to have a thorough understanding of the scenario and the tasks, preventing RL from wandering in a sparse search space.

Furthermore, HRL is inherently well-suited for tasks that involve a progressive logical relationship [37, 38]. A typical work fuses simplified environmental observation and proprioceptive state then generates lower-level control commands for actuators[39]. Some recent attractive benchmarks like HumanoidBench [40], also divide tasks into planning and reaching control and learning hierarchically.

## 3 Problem Statement

We first formulate the problem of learning embodied cooperative behaviour in a swarm of agents, then try to solve this problem with MARL. Since the number of agents is fixed within an episode, the synchronous framework and policy gradient descent algorithms are selected. To realize real-time execution, we further proposed a fully-decentralized training framework to support our implementation. It is crucial to mention that agents in the group might be homogeneous or heterogeneous, and agents in the same species share the same policy.

The embodied cooperation problem can be formulated as a Markov Decision Process (MDP) mathematically expressed as $\langle \mathcal{S}, \mathcal{A}, \mathcal{T}, \mathcal{R} \rangle$ including $N$ agents in $M$ heterogeneous species. $\mathcal{S}, \mathcal{A}$ are state and action tuples, respectively, where $\mathcal{S} = (S_1, S_2, \cdots, S_N)$, $\mathcal{A} = (A_1, A_2, \cdots, A_N)$. $\mathcal{T}$ is the state transition function, and $\mathcal{T}(s'|s, a)$ represents the transition probability distribution of the system when transiting into the next states $s' = (s'_1, s'_2, \cdots, s'_N) \in \mathcal{S}$ from states $s = (s_1, s_2, \cdots, s_N) \in \mathcal{S}$ after taking a set of actions $a = (a_1, a_2, \cdots, a_N) \in \mathcal{A}$. Besides, $\mathcal{R} = (R_1, \cdots, R_M)$ denotes the tuple of reward functions in which different types of agents corre-

spond to distinct reward functions. $r_i = R_i\,(s_i, a_i)$ is the reward given to agent $i$ after taking actions $a_i$ under states $s$.

For the homogenous multiple agents group, the reward function $R$ is determined and shared for each agent. The aim is to find the best shared policy $\pi^*$, to maximize the expected average value of cumulative discount reward:

$$\pi^* = \underset{\pi}{\arg\max} J_\pi = \frac{1}{N} \sum_{i=1}^{N} \sum_{t=0}^{\infty} \gamma^t \cdot \mathbb{E}\left[R_i(s_t^i, a_t^i)|\pi_i\right] \tag{1}$$

where $s_t^i \in S_i$ and $a_t^i \in A_i$ are the state and action of the $i$-th agent. There are some subtle differences in the representation of the heterogeneous population. The aim is to find the best policy tuple $\Omega^* = (\pi_1, \cdots, \pi_M)$ to maximize the joint multiplication of costs $J$:

$$\Omega^* = \underset{\Omega}{\arg\max} \prod_{m=1}^{M} J_m, \text{ where } J_m = \frac{1}{K} \sum_{i=1}^{K} \sum_{t=0}^{\infty} \gamma^t \cdot \mathbb{E}\left[R_m(s_t^i, a_t^i)|\pi_m\right] \tag{2}$$

where $J_m$ denotes the cost function of type $m$ species, and $K$ is the agent number of this species. The final optimization cost for the heterogeneous population is a form of co-multiplication because task assignment exists among the entire team. Each subtask must be completed otherwise the overall task will fail. Even if some subtasks obtain high rewards, any failure in a subtask will prevent the achievement of the overall goal.

## 4 Distributed Hierarchical Reinforcement Learning

As aforementioned, locomotion control for embodied cooperation is an challenging task. There are three main difficulties: (i) *How to decompose and decouple the complicated tasks into a step-by-step mode?* (ii) *Whether the trained models can expand to a large scale of multiagent collaboration?* (iii) *How to establish the temporal and spatial correlation among collective behaviors?* In this part, we will address each of these three questions in detail.

### 4.1 Task Decomposition and Hierarchical Learning

**Network Structure:** Traditionally, a hierarchical neural network structure divides tasks into control and decision layers. Unlike this, in order to address collaborative tasks, we propose a deeper hierarchical structure consisting of three layers. We designate perception-related tasks as the upper layer of the three-layer network that extracts exteroceptive observation including other agents' information into a feature vector; then, we define the middle layer of the network as the planning and decision layer, which provides spatiotemporally correlated commands to the lower-level controller; finally, the lower-layer network focuses on the execution of commands, which can be pre-trained in various modes. The overall information flow in the proposed hierarchical network is illustrated in Fig. 2.

**Hierarchical Perception Processing:** Each layer of the network is responsible for processing distinct types of raw data, which relates to how we acquire and classify perceptual information. Proprioceptive perceptual states including local position, velocity, and joint torque are closely associated with personal locomotion control. Therefore, the proprioceptive state $p$ should only processed by the lower-layer policy. Furthermore, the exteroceptive state $e$ formed by concatenating environmental perception which provides terrain information in the form of a height measurement array [41] with relative observations of other agents [5], as depicted in Fig.3.

### 4.2 Distributed Scalable HRL

In contrast to previous studies that employed HRL or end-to-end approaches for robotic cooperative manipulation or locomotion, our distributed HRL method considers the capability of scaling the population. To achieve this, we need to strictly define the format of input and output information for each agent, particularly the observation. Herein, we propose to use partial observation of

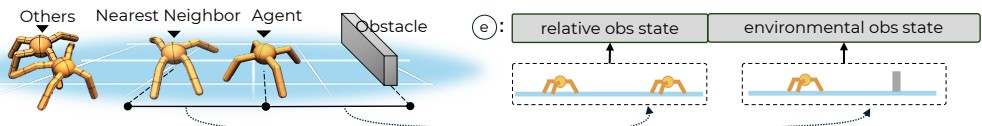

Figure 3: Exteroceptive state configuration in distributed HRL.

agents related to their colleagues, according to their visibility [42]. Each agent can only obtain the nearest neighbour's relative position. As illustrated in Fig.2, the dimension of exteroceptive state $e$ is limited. This actually changes the difficulty of collaboration learning. Since partially observed agents can only see nearby peers and environmental information, they need to estimate the states of other invisible agents from the dynamics of the environment and predict future changes in the environment. However, such observation settings theoretically enable our framework to scale to a significantly large amount of agents system, which is unattainable by CTDE frameworks.

### 4.3 Spatiotemporal Memory Recurrence

Capturing and leveraging spatiotemporal features is crucial for long-term decision-making tasks [43] particularly the complex embodied cooperation tasks we designed. We anticipate that fully-decentralized agents should possess the capability to comprehend their own and other agents' continuous behaviors, thus forming collaborative relationships in space across continuous time intervals. Furthermore, this capability holds particular significance in dynamic environments, especially within fully-decentralized nonstationary multi-agent environments. Therefore, we propose spatiotemporal memory recurrence utilizing RNNs to enhance the agent's ability to make informed decisions over extended periods and complex environments. Consequently, we maintain a hidden recurrent state $h$ to pass the feature of scenarios continuously, as shown in Fig. 2.

### 4.4 Training Curriculum

Acquiring collaborative behaviours requires the agents' random search to span a sufficiently large space, which in complex tasks can often lead to an inability to learn usable knowledge due to the vastness of the search space. In this case, we design some appropriate training curricula that reasonably use sparse rewards and dense rewards in the early stages to intentionally guide the agent to approach success. The curriculum designs are described in detail in the appendix.

## 5 Experiments

The experiments are conducted on a workstation with Intel Xeon W-2150b and RTX2080Ti. Independent Proximal Policy Optimization (IPPO) in a multi-agent setting is employed as the algorithm. Adam optimizer is used with a learning rate of 0.0005. PPO clipping is 0.2, the discount factor is 0.995. We could complete the training within several hours thanks to IsaacSim GPU parallel sampling via multiple environments [4].

### 5.1 Environments Construction

Three scenarios are established that represent different task types: Cooperative Transport, Corridor Crossing and Ravine Bridging.

**Cooperative Transport:** Transporting an object collaboratively is a classical task in the multiagent system field. Herein, we build an embodied cooperative transport task via locomotion control. A group of Ant attempt to move a cylinder object to the red target area, shown in Fig.1(a). We consider fully decentralised and distributed cooperation in a homogeneous and self-organized flock. Thus, the scale of flock can have theoretically unlimited expansion.

**Corridor Crossing:** A group of agents navigate through a narrow corridor sequentially, aiming to complete the passage as quickly as possible. Only one person can pass through the slit at any one time. Agents in this scenario should adjust their position and speed reasonably according to the surrounding observations and pass through a narrow passage without collision. Cooperation in formation is required, as shown in Fig. 1(b). The number of agents can also be expanded in this task.

**Ravine Bridging:** As illustrated in Fig. 1(c), two agents push a movable bridge, enabling another group of agents to cross the chasm smoothly via the bridge. There are two categories of agents in this scenario, which is different from the other two environments. Agents in `Ravine Bridging` are heterogeneous, which means we should maintain more than one policy for different missions.

## 5.2  Results of Distributed HRL

### 5.2.1  Training Basic Locomotion Control

In the training of lower-level motion control, two control paradigms are available: velocity control and position control. We first train the two control paradigms in the single-agent environment to obtain accurate control of the fundamental actions. We give a reciprocal $L_2$ distance between the current position and the target position, or the dot product of the agent velocity and target velocity, as the main reward in each training curriculum. Fig 4(a) shows the training results of two paradigms. After training, We find that the velocity control offers greater flexibility than position control. However, position control presents higher accuracy. Moreover, velocity control makes it difficult to maintain a stationary position at a specific point.

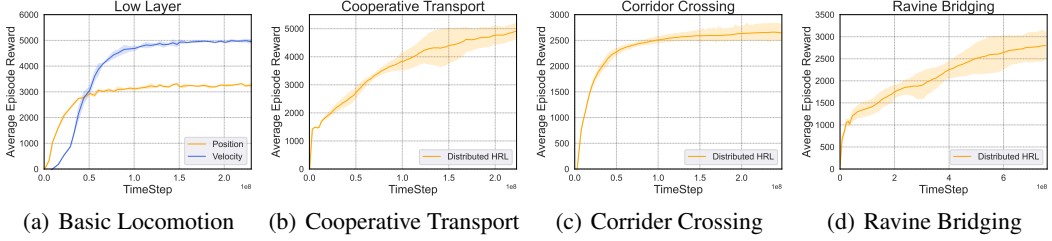

(a) Basic Locomotion    (b) Cooperative Transport    (c) Corrider Crossing    (d) Ravine Bridging

Figure 4: Distributed HRL results for all scenarios.

### 5.2.2  Training the Interactive Collaboration

After the lower layer is elaborately trained,we adopt the position control paradigm and freeze this network. Then, the upper and middle layers are trained on different scenes via distinct reward mechanisms. The training results are shown in Fig. 4(b,c,d).

**Cooperative Transport:** This task reveals clear collaborative behaviors among agents.They demonstrate a strong understanding of the situation, actively choosing tasks based on their circumstances. Some agents first converge towards the object and slightly adjust their position for collaboration; meanwhile, other agents move directly towards the target and prepare to decelerate the object, ensuring it falls within the target area accurately and stably. The success rate of this task reaches approximately 75%.

**Corridor Crossing:** Agents in this task naturally develop characteristics of: "sacrifice" or "concession". If the model pre-trained in a single-agent scenario is directly applied to a multi-agent scenario, all agents would get stuck in the corridor, unable to move. However, after training, agents learn to observe the relative positions of others and adjust their speed accordingly, choosing the appropriate timing for moving or waiting. Typically, agents in the middle rush through the corridor, while others approach the entrance more slowly and accelerate after observing the passage of agents ahead. The success rate of this task reaches approximately 88%.

**Ravine Bridging:** The group in this task is a heterogeneous team. Although all agents are `Ant`, there are two sets of policies: one for agents on the plane and one for agents in the ravine. Agents on the plane will arrive at the edge of ravine and stop, waiting for the agents in the ravine to transport the bridge to the front of them.The success rate of this task reaches approximately 50%. The main reason is that `Ants` in the ravine might get stuck or fall.

Table 1: Success Rate Comparison: We present success rates with standard error on the three tasks, comparing our method to ablations. Each percentage results from 500 trials — 100 attempts per five independently trained models.

| Success rate | HRL | No Hierarchy | No Spatiotemporal Memory |
|---|---|---|---|
| Cooperative Transport | $\mathbf{76.8 \pm 5.3}\%$ | $0.0 \pm 0.0\%$ | $9.6 \pm 3.2\%$ |
| Corridor Crossing | $\mathbf{88.4 \pm 4.8}\%$ | $0.0 \pm 0.0\%$ | $12.0 \pm 4.1\%$ |
| Ravine Bridging | $\mathbf{50.4 \pm 4.4}\%$ | $0.0 \pm 0.0\%$ | $5.4 \pm 4.2\%$ |

## 5.3 Ablation

### 5.3.1 The role of Hierarchical Learning

In this experiment, we replace the hierarchical networks with a single-layer MLP. Given no hierarchical structure, there is no form of target command and velocity command, which precludes the possibility of training an identical lower-level controller in advance. Therefore, when training without hierarchy, we provide the same training curriculum tailored to each task, as comprehensively detailed in the Appendix. Furthermore, we provide proprioceptive states and exteroceptive states as observations and the same task reward and locomotion reward as the total rewards.

We find that agents can only learn simple strategy and get the basic reward which are easy to reach. These results show that hierarchy is crucial to success in more complex tasks that require coordination. Without hierarchy, agents cannot complete the mission (in Table. 1). The reason for failure is obvious. As with other HRL objectives, a single network facing high-dimensional observations and complex multi-item reward functions finds it difficult to distinguish features between channels and learn behaviour guided by a single main reward. Especially, we need a frozen lower-level locomotion controller, which is the foundation for success.

### 5.3.2 The effectiveness of Spatiotemporal Memory

To validate the effectiveness of spatiotemporal features, we re-product the method in previous work [3] as the baseline, which does not consider the event correlation. We provide the same low-level locomotion controller. The approach proposed in [3] which does not have RNNs, converges to a slightly higher reward than in the case of no hierarchy but is still far from completing the mission. As shown in Table. 1, the success rate of this method maintains a relatively low level which rarely exceeding 15%. We find that agents cannot learn a set of systematic behaviours in specific tasks, and there are occasional successes. Actually, if the agents only react to the current observation, they could never form strategic behaviours. In other words, he would never think for himself in the next second without spatiotemporal memory. This is even more deadly in collaborative tasks, as we can obviously find that the conciliatory and sacrificial behaviour emerging in crossing corridor tasks is a strategy of thinking for long-term consequences.

## 5.4 Scalability Experiments

We emphasize the advantages of our proposed fully-decentralized approach in terms of scalability, which includes the advantage of training when the number of agents is large, as we do not need to provide the global observations of all agents like CTDE framework during training to avoid the huge increase in training complexity. On the other hand, our framework also has advantages at the ability of directly transferring the trained model to more agents tasks.

First, we demonstrate the capability of our framework to effectively train agents with nice performance as the number of agents increases, even when these agents only have access to partial observations. We validate this in the task cooperative transport as illustrated in Fig. 5(a). The final reward increments as the number of agents rises, which is also in line with our expectations for this task. The participation of more agents should indeed lead to better transportation results.

Second, we conducted zero-shot transfer tests on our trained model (trained with four agents) across diverse populations of `Ant` in each task(Agent number indicates the agent on the plane in the task Ravine Bridging), and the success rates of each task with different populations are shown in the Tab. 5.4. The outcomes demonstrate the potential of our framework's transfer capability.

| Success rate | Two Ants | Three Ants | Four Ants | Five Ants |
|---|---|---|---|---|
| Cooperative Transport | $12.8 \pm 5.6\%$ | $39.8 \pm 4.5\%$ | $\mathbf{76.8 \pm 5.3}\%$ | $82.2 \pm 4.5\%$ |
| Corridor Crossing | $97.1 \pm 1.8\%$ | $93.6 \pm 2.2\%$ | $\mathbf{88.4 \pm 4.8}\%$ | $73.5 \pm 2.5\%$ |
| Ravine Bridging | $\mathbf{50.4 \pm 4.4}\%$ | $36.2 \pm 7.8\%$ | $22.3 \pm 5.1\%$ | $10.9 \pm 3.3\%$ |

Third, we conduct the experiment to show the advantages of our fully decentralized method training advantage over the CTDE methods when the number of agents increases. The most appropriate task to show is Corridor Crossing because as the number of agents increases, the difficulty of the task Cooperative Transport decreases, while Ravine Bridging is too complex and the completion rate is too low when the number of agents is large. So we add the number of agents to 6 and 7 in corridor crossing. And the Fig. 5(b)(c) shows that the reward of centralized training grows slower and ultimately not converge.

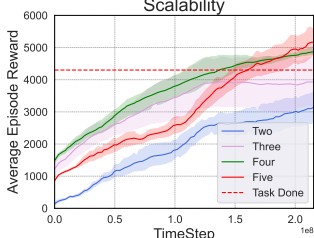 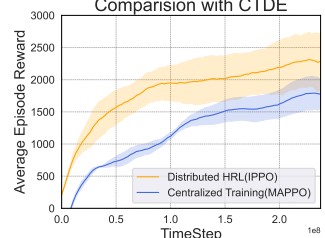 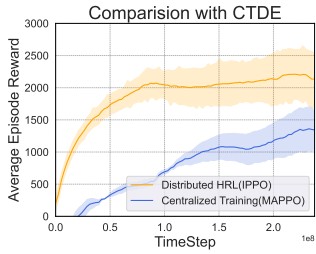

(a) Training results for different `Ant` populations

(b) Training with six agent number

(c) Training with seven agent number

Figure 5: (a) shows the training results of different `Ant` populations used in Cooperative Transport. (b)(c)shows the result training with large agent numbers.

# 6 Conclusion and Limitations

In this work, we propose a distributed HRL architecture for embodied cooperation, and the results showcase the effectiveness of our method in embodied and whole-body control collaborative tasks.

Despite the encouraging findings, it is essential to recognize the limitations of our research. Firstly, although the hierarchical network method endows our framework with a certain ability to address long-horizon problems, its capacity to solve complex issues remains limited as all layers are still fully trained based on reinforcement learning. We primarily relied on continuous reward shaping and increasing the total training steps to obtain better training outcomes. We contend that if we aim to solve more complex long-horizon cooperative tasks with the participation of more intelligent agents, additional planning-based guidance might be necessary at the upper level.

Moreover, our work is indeed more algorithmically and simulation benchmark-oriented, and we have not explored robots with more complex morphology. Despite having already demonstrated the promising results of our method on some tasks and witnessed the emergence of cooperative behavior, there is still a considerable engineering workload required in terms of control and sim2real transfer.

**Acknowledgments**

This work was supported by the National Natural Science Foundation of China under Grants 62025304.

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

# A    Appendix

## A.1    Environment Description

### A.1.1    Cooperative Transport

The agents should cooperate to push a cylinder to a target location. The target position are initialized randomly. The cylinder is set in the middle of the map, and agents are initialized in the four quadrants of the cylinder respectively.

**Observation** The agents receive two sets of observations: (i) a set of "proprioceptive" states which would be given to the lower layer controller. This part of observation is exactly the same in all agents in different tasks because all tasks use the same pre-trained lower layer policy, including their self-observed information, such as its local position, local velocity, angular velocity, orientation, body upright angle, joint force and its limitations, etc. (ii) a set of "exteroceptive" features containing task-relevant information which would be given to upper layer controller. In this task, the observation includes the locations of the nearest agent, the current position of the cylinder, and the destination.

**Rewards** We set three types of rewards: (i) Cylinder distance reward: a term which is the reciprocal L2 distance of the agent to the target. This is the main reward of this task. It will increase when the cylinder becomes closer to the target. The increasing trend is the same as the inverse proportional function. (ii) Agent cylinder distance reward: this term is to encourage agents firstly come closer to the cylinder so that they can learn how to push the object. (iii) Agent distance reward: a term used to guide the agent to walk towards the target, thereby obtaining a large cylinder distance reward.

### A.1.2    Corridor Crossing

The agents are expected to reach a designated line through a narrow slit. Two walls block most of the way forward, leaving only a narrow corridor that only allows one agent to pass at a time. Yet all agents are at the same distance to the entrance of the slit, which means that agents would get stuck at the entrance of the slit together if nobody is slow or stop to wait for others to pass first. We expected agents can acquire a strategy so that all the agents could go through the corridor as fast as possible.

**Observation** The lower layer controllers' observation is the same as the last task. In this task, agents' upper layer controllers' observation includes the locations of itself and the nearest neighbour, the position, translation, and orientation of the walls.

**Rewards** Two rewards are set: (i) Destination reward: if an agent has passed the finish line, all agents will receive a positive reward. It stimulates the strategy allowing as many agents as possible to pass through the corridor to the finish line. (ii) Distance reward to enter: a term to encourage agents first to come close to the corridor.

### A.1.3    Ravine Bridging

The agents are expected to reach a designated line where they must cross over a ravine first. There is a ravine blocking the way of the agents. Agents cannot jump over or climb out of the ravine. Agents need colleagues' help to come over the ravine. At this point, we place a stone of the right size and depth in the ravine to serve as a "bridge" to help the agents cross it. We pre-set two agents on either side of the stone to push the stone to help their colleagues cross the ravine and reach the destination. So this is a complex task that requires teamwork among heterogeneous agents.

**Observation** The observation of the lower layer policy is similar to the previous tasks. Agents' upper layer controllers' observation includes the locations of itself and the nearest agents, the position of the stone, and the position and depth of the ravine.

**Rewards** Two rewards are set: (i) Destination reward: if any agent has passed the finish line, all agents will receive a positive reward. It stimulates the strategy allowing as many agents as possible to go through the ravine and get to the finish line. (ii) Bridging reward: a term to encourage agents in the ravine to carry the stone to the agent above which is the closest to the ravine.

### A.2 Typical Result

#### A.2.1 Cooperative Transport

We analyse the typical result of task cooperative transport. To test the robustness of our method, we conduct more than one hundred trials with randomized reset positions and count the success rate then plot the scatter diagram, as illustrated in Fig. 6(b). Each point represents an initialized position of an agent, and the object is located at (0,0) point. Red points denote agents generated at these positions fail to complete the task, while each blue point denotes a success. We can observe that when the initial position is closer to the object, the success rate tends to be higher, and there is only one failure case within a 5-meter range of the object; conversely, the success rate decreases as the distance from the object increases. This is because the probability of accidents is greatly reduced by short-distance tasks. It is worth mentioning that most of the failure cases result from the agent tipping over or pushing the object obviously to the side with uneven force during the pushing process. Fig.6(c)(d) showcase the evaluation results under no-hierarchy and no-spatiotemporal memory. Only few successes can be found in the evaluation of no-spatiotemporal memory, and all cases fail in training without hierarchy.

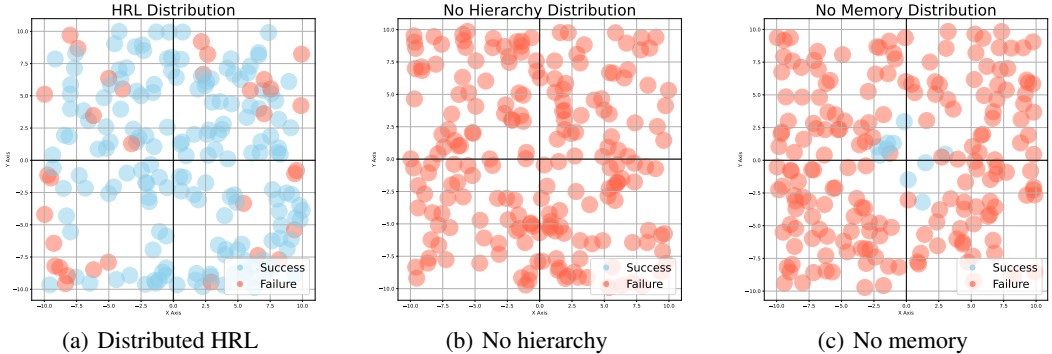

(a) Distributed HRL        (b) No hierarchy        (c) No memory

Figure 6: (a)(b)(c) shows the success and failure distribution of task Cooperative Transport.

#### A.2.2 Corridor Crossing

We record the success rates of each `Ant` at its initial position separately, as illustrated in Fig. A.3. We have a higher success rate when the initial position is closer to the center, and a higher failure rate when the initial position is closer to the edge. This is also explainable, as the `Ant` at the centre is closer to the slit and does not need to adjust velocity.

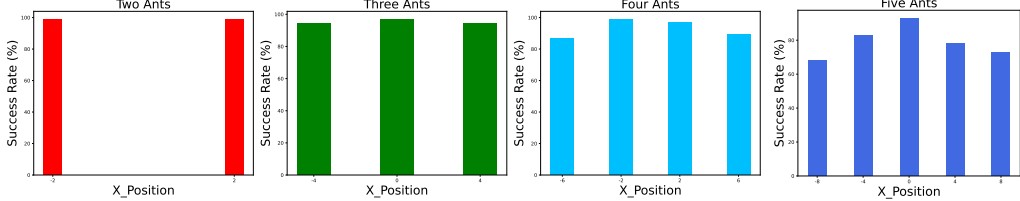

### A.3 Comparison to Other MARL Alogithms

We conduct comparison between the proposed distributed HRL method to other state-of-the-art MARL approaches based on the article [19]. Specifically, we compare our fully-decentralized method with MAPPO and MASAC in task Cooperative Transport(4 agents) since the Isaac Sim algorithm is nested within the RLGame and the CTDE algorithms supported by RLGame are only MAPPO and MASAC. Fig. A.3 shows the results. We find that MAPPO converges a little faster than

distributed method because it is centrally trained with global observations. However, the CTDE algorithm has worse performance and take much more training time when facing a larger number of agents, which has been demonstrated in Fig. 5 of the scalability experiments part.

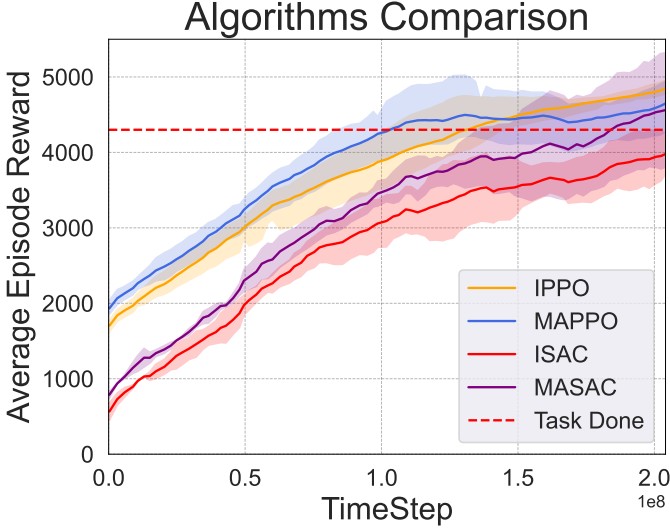

## A.4 Training pipeline of Ravine Bridging

As we mentioned in Appendix A.2, the Ravine Bridging task is more intricate, which means we have to provide more guidance. And the agent's behavior is divided into stages, so the training process must also be divided accordingly. Therefore, there is a series of pre-train courses to complete this task. For the agent in the ravine, its upper layer receives observations that include the position of the bridging stone, its own position, and the position of the Ant on the plane closest to the ravine. The Ant on the plane receives observations that includes the position of the stone, its own position, and the x coordinate of the ravine and the finish line. Before training together, we train them separately first to obtain their featured skills. For the agents in the ravine, we randomize the position of the Ant closest to the ravine in the input and give the agent a y-axis direction distance reward of the distance between the stone and the randomize Ant's position, so that the agent in the ravine has the ability to move the stone to the closest Ant on the plane. For the agents on the plane, we add a reward for the x-axis direction distance to the finish line and a punishment for falling (death), so that it learns the ability to move straight towards the finish line and stop before falling into the ravine. Finally, we train all of them together, and once an Ant crosses the finish line, we give all of them a large sparse reward. For all the reward precise equations mentioned above, we will list them in the following section.

## A.5 Reward Details

### A.5.1 Lower layer

Target paradigm

| Reward Type | Formula |
|---|---|
| Distance to target reward | $5 \times \left( \frac{1}{1+\sqrt{(\text{agent\_pos}[:,0]-\text{target\_pos}[:,0])^2+(\text{agent\_pos}[:,1]-\text{target\_pos}[:,1])^2}} \right)$ |
| Alive reward | 0.5 |
| Up reward | 0.1 |
| Action cost | $0.005 \times$ action cost |
| Energy cost | $0.05 \times$ energy cost |
| Dof at limit cost | $0.1 \times$ dof at limit cost |

Velocity paradigm

| Reward Type | Formula |
|---|---|
| Vel reward | $5 \times \mathrm{clamp}(\mathrm{vel}[:,:2] \cdot \mathrm{cmd\_direction}, \max = 1)$ |
| Alive reward | 0.5 |
| Up reward | 0.1 |
| Action cost | $0.005 \times \mathrm{action\ cost}$ |
| Energy cost | $0.05 \times \mathrm{energy\ cost}$ |
| Dof at limit cost | $0.1 \times \mathrm{dof\ at\ limit\ cost}$ |

### A.5.2 Upper layer

Cooperative Transport

| Reward Type | Formula |
|---|---|
| Cylinder-target distance reward | $10 \times \left( \frac{1}{1+\sqrt{(\mathrm{cylinder\_pos}[:,0]-\mathrm{target\_pos}[:,0])^2+(\mathrm{cylinder\_pos}[:,1]-\mathrm{target\_pos}[:,1])^2}} \right)$ |
| Agent-cylinder distance reward | $5 \times \left( \frac{1}{1+\sqrt{(\mathrm{agent\_pos}[:,0]-\mathrm{cylinder\_pos}[:,0])^2+(\mathrm{agent\_pos}[:,1]-\mathrm{cylinder\_pos}[:,1])^2}} \right)$ |
| Agent-target distance reward | $5 \times \left( \frac{1}{1+\sqrt{(\mathrm{agent\_pos}[:,0]-\mathrm{target\_pos}[:,0])^2+(\mathrm{agent\_pos}[:,1]-\mathrm{target\_pos}[:,1])^2}} \right)$ |
| Reach destination reward | 50 (all agents) |

Cross Corridor

| Reward Type | Formula |
|---|---|
| Reach destination reward | 50 (all agents) |
| Agent-target distance reward | $5 \times \left( \frac{1}{1+\sqrt{(\mathrm{agent\_pos}[:,0]-\mathrm{target\_pos}[:,0])^2+(\mathrm{agent\_pos}[:,1]-\mathrm{target\_pos}[:,1])^2}} \right)$ |

Ravine Bridging

| Reward/Punishment Type | Formula |
|---|---|
| Ravine pretrain: Distance of stone and nearest agent reward | $5 \times \left( \frac{1}{1+|\mathrm{stone\_y}-\mathrm{agent\_y}|} \right)$ |
| Plane pretrain: Distance to destination line reward | $5 \times \left( \frac{1}{1+|\mathrm{destination\_x}-\mathrm{agent\_x}|} \right)$ |
| Plane pretrain: Falling down punishment | -20 |
| Integrated training: Pass destination line reward | 50 (all agents) |

