# OpenReview forum: "Learning a Distributed Hierarchical Locomotion Controller for Embodied Cooperation"
_robot-learning.org/CoRL/2024/Conference — CoRL 2024_

### Official Review · Reviewer_gDjg · 2024-07-20
**Hierarchical RL for Cooperative Locomotion Tasks**

**Originality:** 3
**Technical Quality:** 3
**Clarity Of Presentation:** 2
**Potential Impact:** 3
**Recommendation:** 3
**Confidence:** 5

**Review:**

This work presents a hierarchical reinforcement learning architecture for multi-agent cooperative locomotion tasks. The hierarchical architecture consists of three layers. The first layer is responsible for perception, receiving exteroceptive information, and feature extraction. The second layer handles decision-making and planning, sending commands to the low-level controller. The third layer focuses on executing these commands. This controller is capable of performing cooperative tasks such as Ravine Bridging, Cooperative Transport, and Corridor Crossing. It has been demonstrated that these tasks cannot be successfully completed with a single layer; the hierarchical scheme is essential for success. However, a new policy must be trained for each new task. The paper is well-written but could benefit from improved presentation. Experimental validation or more realistic robotic morphology is needed to demonstrate the algorithm's effectiveness on real robots.

## Major comments
- The proposed tasks, particularly Ravine Bridging, showcase an impressive performance that would be challenging without the hierarchical RL proposed in this paper. It is crucial to clearly state the conditions and information about the Ravine Bridging task, as it is unclear what information is provided to the agents. Additionally, the reward structure for the bridging task should be explained with precise equations. You mention training two separate policies: one for agents on the plane and another for agents in the ravine. Please clearly explain the differences, commonalities, and training phases for these two separate policies. How are they trained—together or separately? Please clarify the training pipeline.

- The current agents are toy ant agents. Experimental validation is a critical missing component of the paper. Either simulating a more realistic morphology, such as Unitree quadrupeds, or conducting experimental validation is necessary to demonstrate how the proposed scheme works on real robots.

- Since the paper is simulation-based and focuses on the methodology, the current supplementary material is insufficient for reproducing the results. Therefore, open-sourcing the code is necessary and would enhance reproducibility and validation.

- Regarding the Cooperative Transport task, the current position of the cylinder is part of the observation space. Is this an absolute value? If so, how do you determine the absolute position of an object with respect to the world frame without ground truth in a real robot experiment?

- The Spatiotemporal Memory effect on the learning task should be explained more simply and clearly.

- The lower-level layer, which is pre-trained, experiences some proprioceptive information during training. When the high-level policy is training, the low-level controller might encounter some previously unseen proprioceptive information (information not experienced during training). How do you ensure this does not happen and that the low-level controller will not encounter out-of-distribution information?

- Adding a subsection to elaborate on the limitations of the work would improve the paper.

## Minor comments:
- For the reward function in the supplementary file, please provide the exact equation.
- Abbreviations such as MARL (multi-agent reinforcement learning) should be fully explained in the video to make it self-explanatory, ensuring the audience understands the abbreviations.
- Adding a narrative can improve the video.
- What are the labels of the x and y axes in Figure 6 (b, c, d)?

**Quality Of The Limitations Section:**

1

**Questions For Rebuttal:**

Please check the above review comments for the questions.

**Robotics Focus:**

3

**Summary Of Paper:**

This work presents a hierarchical reinforcement learning architecture for multi-agent cooperative locomotion tasks. The hierarchical architecture consists of three layers. The first layer is responsible for perception, receiving exteroceptive information, and feature extraction. The second layer handles decision-making and planning, sending commands to the low-level controller. The third layer focuses on executing these commands. This controller is capable of performing cooperative tasks such as Ravine Bridging, Cooperative Transport, and Corridor Crossing. It has been demonstrated that these tasks cannot be successfully completed with a single layer; the hierarchical scheme is essential for success. However, a new policy must be trained for each new task.

**Summary Of Recommendation:**

The methodology of the paper and the simulation results are nice. However, the presentation could be improved, and validation using a real robot or a realistic robot morphology in the simulation is necessary. Therefore, I recommend a weak acceptance.

---

### Official Review · Reviewer_Mqnz · 2024-07-21

**Originality:** 3
**Technical Quality:** 2
**Clarity Of Presentation:** 2
**Potential Impact:** 3
**Recommendation:** 3
**Confidence:** 5

**Review:**

# Strengths
* The creation of benchmark environments in IsaacSim provides a platform for future research of cooperative multi-agent locomotion and manipulation.
* The proposed method outperforms the baseline and its ablations by a large margin.
* The idea of the paper is straightforward and easy to understand.

# Weaknesses
* The authors should consider comparing to more recent works, such as the one below, which also proposes a hierarchical decentralized multi-agent reinforcement learning framework for cooperative navigation and object transport:
    * Solving Multi-Entity Robotic Problems Using Permutation Invariant Neural Network. An et al., arXiv 2024.
* The method is not validated on real robots.
* The high-level and medium-level hierarchies are actually acted as different layers in a neural net but not separate decision hierarchies. The utilization of RNN is kind of standard in most current RL frameworks for partially observable environments.
* The problem formulation of this paper should be improved. The authors use IPPO in the experiment, which is fully decentralized and does not belong to the centralized training decentralized execution (CTDE) framework.
* It seems that the cooperative transport task can be well done by two agent (from video), and there are idle agents if more than two. Adding number of agents does not effectively show the scalability of the method.

**Quality Of The Limitations Section:**

1

**Questions For Rebuttal:**

* How many seeds are used to demonstrate the results?
* Why can the method with noisy hidden states sometimes outperform the baseline without using hidden states? What are the type and level of noises applied?
* The authors mention that the proposed method has great potential to migrate a trained policy to a larger number of agents. Does any results support this argument?

**Robotics Focus:**

3

**Summary Of Paper:**

This paper proposes a decentralized hierarchical structure for multi-robot whole-body locomotion and manipulation. The high-level layer is an observation encoder, the medium-layer is a hidden state encoder, i.e., RNN, and the low-level hierarchy is a pre-trained locomotion controller. The proposed method is validated on three multi-agent cooperative tasks built in IssacSim compared to one baseline.

**Summary Of Recommendation:**

I believe the presentation of this paper still needs to be improved and its contributions and novelties have not been sufficiently elaborated.

---

### Official Review · Reviewer_pYHV · 2024-07-23
**Some additional information and evidence needed**

**Originality:** 3
**Technical Quality:** 3
**Clarity Of Presentation:** 4
**Potential Impact:** 3
**Recommendation:** 3
**Confidence:** 3

**Review:**

The paper addresses a relevant and challenging problem in robotics: learning cooperation in multi-agent systems. The proposed distributed HRL architecture is well-motivated and provides a structured approach to decompose complex tasks. The experiments on the constructed benchmark environments provide some insights into the effectiveness of the approach. However, without hardware experiments, the “embodied” cooperation skill is not demonstrated.

Strengths:

1. The distributed HRL framework, incorporating spatio-temporal memory is a well-motivated network architecture for MARL.
2. The method demonstrates the potential for scaling to larger numbers of agents by focusing only on nearest neighbor agents.
3. The experiments showcase the emergence of some cooperative behaviors, such as task allocation and coordination.
4. The authors' construction of benchmark environments provides a valuable resource for MARL evaluation.
5. The paper is well-written and clearly presents the methodology and experimental results.

Weaknesses:

1. While claiming applicability to heterogeneous groups, the scalability evaluation is limited to the homogeneous setting.
2. A broader comparison to state-of-the-art MARL methods would help to better contextualize the contribution and performance of the proposed method.
3. The evaluation is confined to simulation, and real-world applicability remains to be demonstrated.
4. The paper needs a detailed limitations section.

**Quality Of The Limitations Section:**

1

**Questions For Rebuttal:**

1. The authors claim the method is scalable and can be applied to both homogeneous and heterogeneous groups. However, the experiments only demonstrate scalability within the homogeneous setting (Cooperative Transport). Can the authors provide additional results or analysis to support the claim of generalizability to heterogeneous groups? The scalability result considers up to 5 agents. Can it be demonstrated that baseline methods cannot scale to 5 agents?
2. While the paper provides a baseline comparison to a previous work [3], it would be beneficial to compare the proposed distributed HRL method to other state-of-the-art MARL approaches to better contextualize the contribution and performance of the proposed method. Specifically, consider running this method on “BenchMARL: Benchmarking Multi-Agent Reinforcement Learning” (https://arxiv.org/pdf/2312.01472) to get a comparison against many MARL algorithms.
3. The current evaluation is conducted in simulation. Could the authors discuss the potential challenges and steps involved in evaluating their method on real-world robots?
4. In section 5.3.1, when discussing the role of hierarchy, the baseline network should be matched with the hierarchical network in the number of parameters for fair comparison. Please mention the network sizes. There should also be a comparison with a baseline RNN model without a hierarchy of similar size.

**Robotics Focus:**

3

**Summary Of Paper:**

This paper presents a distributed hierarchical reinforcement learning (HRL) approach for multi-agent cooperation tasks. The policy network is a three-layer hierarchical structure: feature extraction from exteroceptive observations from environment and neighboring agents, spatiotemporal decision-making using recurrent neural networks (RNNs), and low-level action control. The method is evaluated on three custom-built environments within IsaacSim/Gym. The experimental results demonstrate the emergence of cooperative behaviors.

**Summary Of Recommendation:**

The paper presents a sound approach to MARL systems and some additional experimental results will strengthen the claims in the paper.

---

### Author Rebuttal · Authors · 2024-08-06

We are very grateful to all reviewers for their detailed and useful feedback. We have revised the paper based on the feedback and answered all the questions raised by the reviewers. We have submitted the revised paper in the rebuttal files. Here, we list the main adjustment and supplement to the paper:
1. More comparative experiments. We have compared our fully decentralized training algorithms with more CTDE algorithms. And including the special case to illustrate the results when agent numbers are larger.
2. We put the results of scalabilty experiments on the main paper and provide more experiments and analysis in this part.
3. A detailed explanation about the environment and training rewards. Especially the training pipeline for complex tasks.
4. A limitation part to clarify the problems encountered in our work and the constraint of our algorithm.
5. We further clarify the the advantages of our fully-decentralized method over the general CTDE framework and the significance of spatiotemporal memory in our paper.
If the reviewers still have remaining questions after reading the revision and responses, we are happy to provide additional responses.

---

### Decision · Program_Chairs · 2024-09-05

**Decision:**

Accept

**Comment:**

The reviewers are relatively positive but not enthusiastic about the work. The reviewers highlight that the proposed approach and experiments are well motivated. The authors proposed design decisions for the network architecture / composition and observations that make sense for the task. The main concern of the reviewers is the thoroughness of the evaluation  the limited to non-existent baselines and the evaluation in simulation using unrealistic embodiment.   I personally want to add that the exact details of the ablation studies are not clear. For example what does no-hierarchy mean? Does this mean no pre-trained locomotion model in which case the comparison would be unfair. Furthermore  it is unclear whether the comparison is really clear as I would expect the baselines to solve at least the corridor task.  We encourage the authors to to better clarify the algorithm and experiments. Further, evaluation in real-world can strengthen the paper and is highly encouraged.